# A Theoretical Solution for Pile-Supported Embankment with a Conical Pile-Head

**Chengfu Zhang, Minghua Zhao \*, Shuai Zhou and Zeyu Xu**

Institute of Geotechnical Engineering, Hunan University, Changsha 410082, China
\* Correspondence: xuzeyu@hnu.edu.cn

**Abstract:** This paper, with a focuses on the pile-supported embankment with a conical pile-head, proposes a theoretical solution which incorporates all the load transfer mechanisms, namely the soil arching effect, the pile–soil interaction, and the support from the substratum, whilst an improved cylindrical unit cell model is introduced to analyze the soil arching effect. The theoretical solution has been verified via numerical analysis and a literature method. The comparative results indicate that the proposed theoretical solution can effectively evaluate the pile-supported embankment with a conical pile-head. Furthermore, parametric studies have also been conducted to analyze the effect of model parameters on the load sharing ratio ($n_e$), the pile–soil stress ratio ($n$), and the pile shaft friction.

**Keywords:** Theoretical solution; soil arching; pile–soil interaction; pile–soil stress ratio; embankment; axisymmetric model

## 1. Introduction

Pile inclusion is considered as one of the most versatile and cost-effective soft soil improvement techniques for its distinct advantages of high construction speed, small total/differential settlement, and low construction cost [1–7]. This technique consists of a grid of piles driven through the soft layer and embedded in an appropriate substratum or the bedrock with an embankment placement above the pile, as shown in Figure 1.

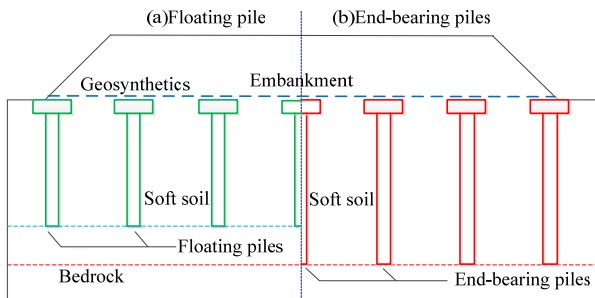

**Figure 1.** Types of pile-supported embankments: (**a**) End-bearing pile (**b**) Floating pile.

The embankment load is transferred to substratum via the pile. To strengthen the effect of the pile, the pile head area should be increased while the pile shaft area remains unchanged. A common technique is to set square or circular concrete slabs on the pile top [8,9]. However, the major limitation of this technique is that it necessitates two separate processes (completion of the pile and installation of the precast slab) and is not easy to carry out [10]. Therefore, a circular pile with a conical head, a type of the variable section pile, can be designed under the concept of the variable section pile. The pile can be constructed at one time, which secures the continuity.



Compared with the equal section pile, the pile with a cone cap (a type of variable section pile) can convert partial vertical load into horizontal load due to its unique geometry. This characteristic can effectively reduce the negative friction and improve the bearing capacity of the composite foundation. Robinsky [11], Zil'berberg [12] have conducted a series of tests on the tapered pile (a type of variable section pile). According to their experiment results, it indicates that the bearing capacity of the tapered pile is far superior to that of the equal section pile. Ladanyi [13] carried out load tests on several types of piles in permanently frozen soil, such as the tapered pile and the equal section pile. Based on the experiment results, we can see that the bearing capacity of the tapered pile is far superior to that of the other types of pile. Sawaguchi [14] also has conducted a series of model tests on the pile with a tapered cap in sandy soil, with the conclusion that little negative skin friction is generated within the tapered cap zone. The circular pile with a conical head, a type of variable section pile, has been successfully applied in a pile-supported embankment near Bourgoin-Jallieu (France). As for this, Dias [15] made an introduction about the construction of a circular pile with a conical head and conducted a numerical study. In summary, the researches on the variable section piles are carried out on the bases of experiments and numerical analysis, and rarely on theoretical analysis.

Generally, the theoretical analysis of the pile-supported embankment includes two calculation steps: one is the soil arching effect; the other is the pile–soil interaction. Theoretically, the existing soil arching models can be roughly classified into three groups [4,8,16]. The first is called the rigid arch model, such as the enhanced arching models [17] and Scandinavian models [18–21]. It is assumed in the model that an arch of a fixed shape can be formed. The load above the soil arch is delivered directly to the pile and the weight below the soil arch is carried by the soft soil. However, in this model, it fails to consider the influence of some physical properties of the fill, such as the friction angle and the compactness. The second is called the limit equilibrium model, such as the Hewlett and Randolph model [22] and the concentric arch model [8]. In the model, it is assumed that the height of the soil arch is 0.5 times the net spacing of pile and the dome or arch feet is in an ultimate stress state. Another is the frictional model [9,23–25], which assumes that the friction exists in the vertical plane along the edge of the pile cap and the equilibrium of the soil is considered based on the Terzaghi [26] theory. However, there is a limitation in the frictional model, where the friction at the plane of equal settlement is not zero, which deviates from the concept of the equal settlement plane.

The mechanisms contributing to the load transfer in the embankment have been the focus of extensive research. However, less attention has been paid to the role of pile–soil interaction and the impact on soil arching [10]. Balaam et al. [27] have presented a finite element method to determine the settlement of granular column-reinforced ground subjected to uniform vertical stress imposed by an embankment. Alamgir [28] has assumed a deformed shape function to simulate the uneven deformation of soft soil and deduced the friction between pile and soil, but has not considered the soil arching effect. Chen [23] has analyzed the pile–soil interaction under the soil arching effect by assuming the distribution equation of pile shaft friction along the length of the pile. Based on Alamgir's analysis, Zhao [29] has considered the impact of the soil arching on pile–soil interaction.

This study puts forward a theoretical method for the pile-supported embankments with a conical head, which accounts for all relevant load transfer mechanisms in a pile-supported embankment, namely, the improved frictional mode, the pile–soil interface friction, and the support of substratum. The solution of the proposed method is acquired by coupling the deformation equation of the embankment with the deformation equation of the composite foundation according to the continuity of stress and displacement. Three key improvements are included in this method. First, the proposed method can be used to evaluate the pile-supported embankment with a conical head. Second, the critical height of the soil arch is not pre-assumed, but is calculated under the deformation continuity condition. Last, in the improved frictional soil arching model, the effect of pile–soil interaction on the development of soil arching is considered.

## 2. Theoretical Model

In Figure 2a, a cylindrical unit cell model is utilized to analyze the load transfer mechanism of the pile-supported embankment. The model is presented by Alamgir (1996) [28]. As described in literature by Chen et al. and Zhao et al. [23,29], the fill is hypothetically divided into two parts: one is an inner cylinder with diameter $d_c$, the other is an outer hollow cylinder with an outer diameter $d_e$ and inner diameter $d_c$. In the derivations, the fill is considered to be homogeneous, isotropic, and non-cohesive with internal friction angle $\varphi_f$, unit weight $\gamma_f$, and Young's modulus $E_f$. Only the vertical deformation is considered for the fill, the soft soil, and the pile. In the inner cylinder, the vertical stress and vertical deformation are uniform at any a certain cross-section. The same assumption is made in the outer hollow cylinder and the pile. However, soil properties are inhomogeneous by nature [30–32], which is the limitation of this model.

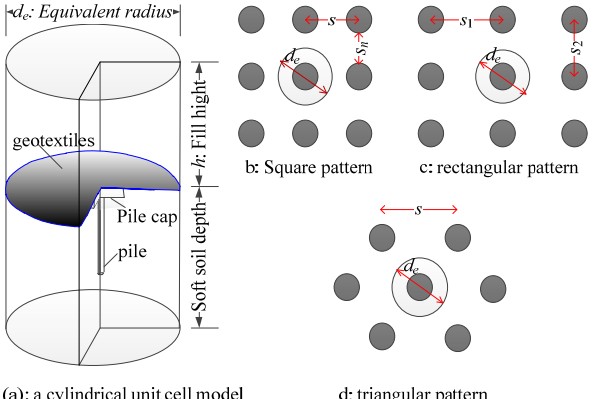

**Figure 2.** The installation pattern of piles: (**a**) a cylindrical unit cell model, (**b**) square pattern, (**c**) rectangular pattern, (**d**) triangular pattern.

Generally, the arrangement of piles is divided into three types, as shown in Figure 2. The axis to axis spacing and net spacing between the adjacent piles is respectively denoted as $s$ and $s_n$. Under the principle of area equivalence, the relationship between $d_e$ and $s$ can be denoted as

$$
\begin{cases}
d_e = 1.05s & \text{triangular} \\
d_e = 1.13\sqrt{s_1 s_2} & \text{rectangular} \\
d_e = 1.13s & \text{square}
\end{cases}
\tag{1}
$$

where $d_e$ is the effective reinforcement diameter of pile; $s_1$ and $s_2$ represent the length and width of the rectangle respectively; $s$ is the axis to axis spacing of the pile.

### 2.1. Soil Arching in the Embankment

As shown in Figure 3, a local coordinate system is established where the origin is situated at the surface of the embankment and the downwards is the positive direction of the $z^*$. The pile is rigid relative to soft soil. The differential settlement occurs between the inner cylinder and the outer hollow cylinder under the load of fill. Therefore, friction is generated on the interface and acts downwards on the inner cylinder but upwards on the outer hollow cylinder. The friction can be acquired by the following equation:

$$
f = \beta \sigma_i(z^*) k_{ae} \tan \varphi_f
\tag{2}
$$

where $f$ is the friction between the inner cylinder and the outer hollow cylinder; $\beta$ is the mobilization coefficient of friction; $\sigma_i(z^*)$ denotes the vertical stress in the inner cylinder; $k_{ae}$ represents active pressure coefficient in embankment; and $\varphi_f$ is the internal friction angle of the fill.

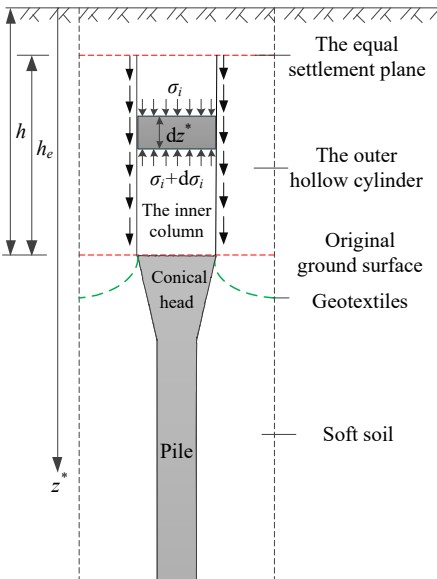

**Figure 3.** The cylindrical unit cell model.

For the sake of simplicity, $\beta$ is usually assumed to be 1 [23,26]. It does not satisfy the stress condition that the friction between the inner cylinder and the outer hollow cylinder ought to zero at the equal settlement plane. In order to satisfy the stress condition and consider the influence of relative displacement, the author assumes that $\beta$ varies linearly from 0 at the equal settlement plane to 1 at the pile top. Considering the flatness of the road surface, the case of $h < h_e$ is beyond the scope of this paper.

$$k = \beta k_{ae} = \begin{cases} 0 & 0 < z^* \leq h - h_e \\ \frac{k_{ae}}{h_e}(z^* + h_e - h) & h - h_e < z^* \leq h \end{cases} \tag{3}$$

where $h$ is the height of embankment, and $h_e$ is the height of the equal settlement plane.

As Figure 3 shows, the inner cylinder is divided into small pieces with a thickness of $dz^*$, and the equilibrium of vertical force can be expressed as

$$S_i\sigma_i(z^*) + \gamma_f S_i dz^* + \pi D f dz^* = S_i(\sigma_i(z^*) + d\sigma_i(z^*)) \tag{4}$$

$$f = \sigma_i(z^*)k\tan\varphi_f \tag{5}$$

where $S_i$ is the cross-sectional area of the inner cylinder, $\gamma_f$ is the unit weight of the fill, and $D$ is the maximum diameter of the conical head.

When $z^* = h - h_e$, $\sigma_i(h - h_e) = \gamma_f(h - h_e)$, Equations (3) and (5) are substituted into Equation (4), then Equation (4) is integrated from $z^* = h - h_e$ to $z^*$, the following formula can be obtained:

$$\sigma_i(z^*) = \gamma_f(h - h_e)H + \frac{H\sqrt{\frac{\pi}{2}}\gamma_f Erf\left[\frac{\sqrt{T}(z^* + h_e - h)}{\sqrt{2}}\right]}{\sqrt{T}} \tag{6}$$

where $T$ and $H$ are the intermediate variables: $T = \frac{4\tan(\varphi_f)k_{ae}}{Dh_e}$, $H = \exp(T\frac{(h_e + z^* - h)^2}{2})$.

At any horizontal sections in the embankment, the equilibrium of vertical force can be denoted as follows:

$$m\sigma_i(z^*) + (1 - m)\sigma_o(z^*) = \gamma_f z^* \tag{7}$$

where $m$ is the area replacement rate, $m = S_i/(S_i + S_o)$, $\sigma_o(z^*)$ is the vertical stress in the outer hollow cylinder, $S_0$ is the cross section area of the outer hollow cylinder.

The vertical stress $\sigma_o(z^*)$ can be formulated as

$$\sigma_o(z^*) = \frac{\gamma_f z^* - m\sigma_i(z^*)}{1-m} \tag{8}$$

The pile–soil stress ratio ($n$) and the load sharing ratio ($n_e$) are defined as follows, respectively:

$$n = \frac{\sigma_i(h)}{\sigma_o(h)} \tag{9}$$

$$n_e = \frac{\sigma_i(h)S_i}{\gamma_f h(S_i + S_o)} \tag{10}$$

The relative displacement incurs the friction. The outer hollow cylinder is subjected to an upward friction, which causes the outer hollow cylinder rebound. In contrast, a compression deformation is generated on the inner cylinder. Therefore, the differential settlement ($\Delta_s$) at the plane of pile top is expressed as

$$\Delta_s = \int_{h-h_e}^{h} \frac{\sigma_i(z^*) - \sigma_o(z^*)}{E_f} dz^* \tag{11}$$

where $E_f$ is the elastic modulus.

## 2.2. Load Transfers between Pile and Soil

Figure 4 illustrates the schematic diagram of load transfer in the reinforced zone. The geometric parameters $L_p$, $r_1$, $r_2$, $\alpha$, and $L_0$ respectively represent the pile length, maximum radius, minimum radius, the cone angle, and the height of the conical head. The neutral point is located where the settlement of the soil and pile is equal and the friction is zero at the neutral point. $L_1$ represents the height from the pile top to the neutral point. There is a local coordinate system that the datum point is located at the pile top and the downward direction is positive. According to the characteristics of the pile–soil interaction, the reinforced zone can be divided into three zones: (1) the zone of the conical head, (2) the zone of positive friction, and (3) the zone of negative friction.

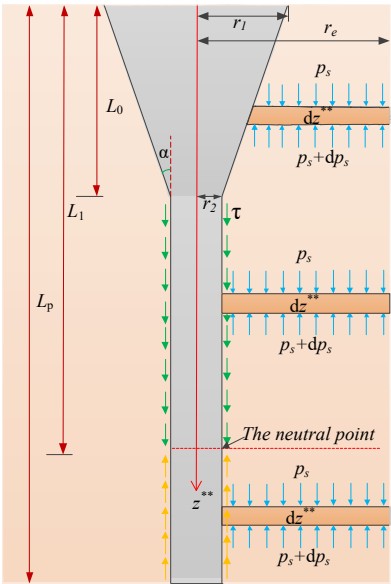

**Figure 4.** Load transfer model of the reinforced zone.

### 2.2.1. The zone of the conical head

The interaction between the conical head and soil is analyzed in this section. As depicted in Figure 5, the interaction force consists of a normal force vertical to the contact surface and friction parallel to the contact surface. The equilibrium equations can be acquired by decomposing the normal force and the friction along the coordinate direction.

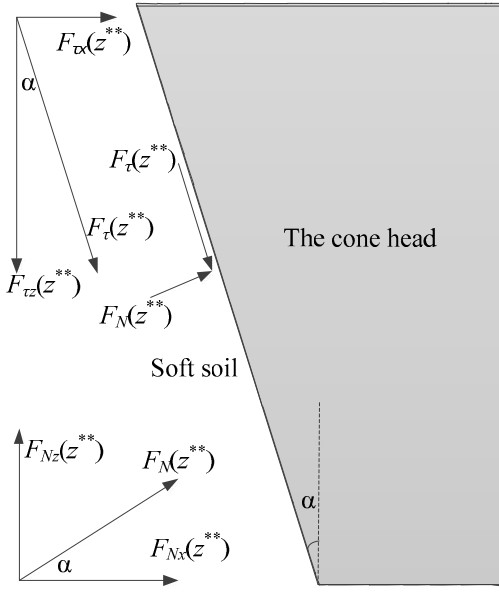

**Figure 5.** Interaction between the conical head and the soil.

Horizontally:

$$F_N(z^{**})\cos(\alpha) + F_N(z^{**})\tan(\varphi_s)\sin\alpha = P_s(z^{**})k_p \tag{12}$$

Vertically:

$$F_N(z^{**})\sin(\alpha) - F_N(z^{**})\tan(\varphi_s)\cos(\alpha) = F(z^{**}) \tag{13}$$

where $F_N(z^{**})$ is the normal stress; $P_s(z^{**})$ is the vertical stress of the soft soil; $F(z^{**})$ is the vertical component of the interaction stress; $\varphi_s$ is the internal friction angle of the soil; and $\alpha$ is the cone angle of the conical head.

By associating Equation (12) with Equation (13), the following equation can be obtained:

$$F(z^{**}) = \frac{Mk_p}{C}P_s(z^{**}) \tag{14}$$

where $M$ and $C$ represent the intermediate variables without physical significance, $M = \sin(\alpha) - \tan(\varphi_s)\cos(\alpha)$, $C = \cos(\alpha) + \tan(\varphi_s)\sin(\alpha)$.

As shown in Figure 4, the surrounding soil within the conical head zone is divided into small pieces with thickness of d$z$. The vertical force equilibrium can be denoted as

$$P_s(z^{**})A_{su} + F(z^{**})A_{b0}\mathrm{d}z^{**} = P_s(z^{**})A_{sd} + \mathrm{d}P_s(z^{**})A_{sd} \tag{15}$$

where $A_{su}$ and $A_{sd}$ respectively represent the upper and lower surface area of the soil element, $Asu = Asd = \pi(re2 - (r1 - k1z^{**})2)$; $A_{b0}$ denotes the perimeter of the soil element, $A_{b0} = 2\pi(r_1 - k_1z^{**})$ and $k1 = tan(\alpha)$.

Substituting Equation (14) into Equation (15) and integrating Equation (15), while introducing the stress continuity condition $Ps(z^{**} = 0) = \sigma o(z^* = h)$, the following formula thus can be obtained:

$$P_s(z^{**}) = [(r_1 - r_e)(r_1 + r_e)]^{\frac{Dkp}{Ck_1}} [(r_1 - r_e - k_1 z^{**})(r_1 + r_e - k_1 z^{**})]\sigma_o(h) \tag{16}$$

The sum of the vertical forces acting on the pile and the soil is equal to the gravity.

$$m_1 p_p(z^{**}) + (1 - m_1)p_s(z^{**}) = \gamma_f h + \gamma_s z^{**} \tag{17}$$

where $m_1$ is the ratio of the conical head area at any horizontal cross section to the whole cylinder area and $m_1 = (r_1 - k_1 z^{**})/r_e^2$; $\gamma_s$ is the unit weight of the soft soil.

Substituting Equation (16) into Equation (17), the stress formula of pile therein reads as follows:

$$
\begin{aligned}
p_p(z^{**}) = &\frac{r_e^2}{(r_1 - k_1 z^{**})^2}[(h\gamma_f + z^{**}\gamma_s) \\
&- [(r_1 - r_e)(r_1 + r_e)]^{\frac{Dkp}{Ck_1}} [(r_1 - r_e - k_1 z^{**})(r_1 + r_e - k_1 z^{**})](1 - \frac{(r_1 - k_1 z^{**})^2}{r_e^2})\sigma_o(h)]
\end{aligned}
\tag{18}
$$

In the conical head zone, the vertical stress on the pile and soft soil can be obtained from Equations (16) and (18). Therefore, the deformation of the soil and pile can be formulated as follows:

$$S_{s1} = \int_0^{L_0} \frac{p_s(z^{**})}{E_s} \mathrm{d}z^{**} \tag{19}$$

$$S_{p1} = \int_0^{L_0} \frac{p_p(z^{**})}{E_p} \mathrm{d}z^{**} \tag{20}$$

### 2.2.2. The Negative Friction Zone

As shown in Figure 4, since the soil moves down relative to the pile in the negative friction zone, the pile is subjected to a downward friction. The Randolph method [33] is adopted to calculate the friction, and it can be shown as follows:

$$\tau = \begin{cases} -\mu_1 k_{as} p_s(z^{**}) & L_0 < z^{**} < L_1 \\ \mu_2 k_{as} p_s(z^{**}) & L_1 < z^{**} < L_P \end{cases} \tag{21}$$

where $k_{as}$ is the active earth pressure coefficient and $k_{as} = \tan^2(45 - \varphi_s/2)$. $\mu_1$ and $\mu_2$ respectively stand for the negative and positive friction coefficients; Giroud [34] suggested the two values should be 0.3.

Similarly, the analysis is carried out using a soil unit with a thickness of $\mathrm{d}z^{**}$. The equilibrium of the vertical force can be expressed as

$$p_s(z^{**})A_s = p_s(z^{**})A_s + \mathrm{d}p_s(z^{**})A_s + \tau A_{b1} \mathrm{d}z^{**} \tag{22}$$

where the soil area around the pile is referred as $A_s = \pi(r_e^2 - r_2^2)$, and $A_{b1}$ is the perimeter of the pile.

Substituting Equation (21) into Equation (22) and integrating Equation (22), while introducing the stress continuity condition at $z^{**} = L_0$, the following formula is obtained:

$$p_s(z^{**}) = e^{\left(\frac{2\mu_1 k_{as} r_2 (z^{**} - L_0)}{r_2^2 - r_e^2}\right)} [(r_1 - r_2)(r_1 + r_2)]^{-\frac{Dkp}{Ck_1}} [(r_1 - k_1 L_0)^2 - r_e^2]^{\frac{Dkp}{Ck_1}} \sigma_o(h) \tag{23}$$

Similarly, the sum of the vertical stress on the pile and soil is equal to the gravity.

$$m_2 p_p(z^{**}) + (1 - m_2) p_s(z^{**}) = \gamma_c h + \gamma_s z^{**} \tag{24}$$

where $m_2$ is the area replacement rate at the lower side of the conical head, and $m_2 = r_2{}^2 / r_e{}^2$.

Equation (25) could be deduced by substituting Equation (23) into Equation (24):

$$p_p(z^{**}) = \frac{r_e^2}{r_2^2} [(\gamma_c h + \gamma_s z^{**})$$
$$- e^{\frac{2\mu_1 k_{as} r_2 (z^{**} - L_0)}{r_2^2 - r_e^2}} \left(1 - \frac{r_2^2}{r_e^2}\right) ((r_1 - r_e)(r_1 + r_e))^{\frac{-Dk_p}{Ck_1}} ((r_1 - k_1 L_0)^2 - r_e^2)^{\frac{Dk_p}{Ck_1}} \sigma_o(h)] \tag{25}$$

In the negative friction zone, the vertical stress of piles and soft soil can be obtained by Equation (25) and Equation (23), respectively. Analogically, the deformation of the soil and pile can be obtained.

$$S_{s2} = \int_{L_0}^{L_1} \frac{p_s(z^{**})}{E_s} dz^{**} \tag{26}$$

$$S_{p2} = \int_{L_0}^{L_1} \frac{p_p(z^{**})}{E_p} dz^{**} \tag{27}$$

### 2.2.3. The Positive Friction Zone

In this section, the pile moves downward relative to the soil and is subjected to an upward friction. Similar to the previous analysis process, the stresses and deformation in the negative friction zone can be acquired.

$$p_s(z^{**})A_s + \tau A_{b1} dz^{**} = p_s(z^{**})A_s + dp_s(z^{**})A_s \tag{28}$$

$$p_s(z^{**}) = e^{\frac{2k_{as} r_2 (z^{**} \mu_1 + L_1(\mu_1 - \mu_2) - L_0 \mu_1)}{r_2^2 - r_e^2}} [(r_1 - r_e)(r_1 + r_e)]^{\frac{-Dk_p}{Ck_1}} [(r_1 - k_1 L_0)^2 - r_e^2]^{\frac{Dk_p}{Ck_1}} \sigma_o(h) \tag{29}$$

$$p_p(z^{**}) = \frac{r_e^2}{r_2^2} [(h\gamma_c + z^{**}\gamma_s)$$
$$- e^{\frac{2k_{as} r_2 (z^{**} \mu_2 + L_1(\mu_1 - \mu_2) - L_0 \mu_1)}{r_2^2 - r_e^2}} \left(1 - \frac{r_2^2}{r_e^2}\right) ((r_1 - r_e)(r_1 + r_e))^{\frac{-Dk_p}{Ck_1}} ((r_1 - k_1 L_0)^2 - r_e^2)^{\frac{Dk_p}{Ck_1}} \sigma_o(h)] \tag{30}$$

$$S_{s3} = \int_{L_1}^{L_p} \frac{p_s(z^{**})}{E_s} dz^{**} \tag{31}$$

$$S_{p3} = \int_{L_1}^{L_p} \frac{p_p(z^{**})}{E_p} dz^{**} \tag{32}$$

### 2.3. Pile toe displacement

In this section, the relative displacement of pile toe to the subsoil was analyzed. The subsoil was assumed to meet the conditions of the Winkler foundation model. Therefore, the relative displacement can be calculated as follows:

$$\Delta_2 = \frac{p_p(L_p) - p_s(L_p)}{k_{bw}} \tag{33}$$

$$k_{bw} = \frac{4G}{\pi r_0 \rho (1 - v)} \tag{34}$$

where $k_{bw}$ refers to the stiffness of the subsoil below the pile toe; $G$ and $v$ stand for the shear modulus and the Poisson's ratio of the subsoil; and $\rho$, the depth impact factor at the pile toe, is 0.85 as recommended by Randolph [33].

### 2.4. Solution

Above the neutral point, the soil moves downwards relative to the pile while the differential settlement amounts to the maximum at the pile top. Below the neutral point, the pile moves downwards relative to the soil and reaches a maximum at the pile toe. Therefore, the relative displacement at the pile top and pile toe can be respectively denoted as $\Delta_u$ and $\Delta_d$.

$$\Delta_u = S_{s1} + S_{s2} - S_{p1} - S_{P2} \tag{35}$$

$$\Delta_d = S_{p3}S_{s3} \tag{36}$$

According to the displacement continuous condition at the pile top and the pile toe, the displacement relationship runs as follows:

$$\Delta_u = \Delta_s \tag{37}$$

$$\Delta_d = \Delta_2 \tag{38}$$

Substituting Equation (19)–(20), (26)–(27) into Equation (35) comes to $\Delta_u$, and substituting Equation (31) and Equation (32) into the Equation (36) comes to $\Delta_d$. $\Delta_u$ and $\Delta_d$ are a function of $L_1$ and $h_e$. Similarly, $\Delta_s$ and $\Delta_2$ are also a function of $L_1$ and $h_e$. Thus, Equation (37) and Equation (38) are the set of equations about $L_1$ and $h_e$. The value of $L_1$ and $h_e$ are obtained by solving the equation set and the value of other variables can be obtained accordingly.

## 3. Validation

### 3.1. Validation 1

In this section, a numerical model is built in FLAC-3D. The numerical model aims to better understand the performance of the pile-supported embankment with a conical head and verify the theoretical solution. The foundation soil consists of three layers. The first layer is an artificial working platform with a thickness of 2 m, which is the pre-treatment layer for carrying the machinery load during construction. This is equivalent to the uppermost clayey crust with relatively high plasticity. The second layer is a soft soil layer with a thickness of 10 m. The third layer is a hard soil layer with a thickness of 6 m. The pile penetrates into the soft soil layer until the toe reaches the hard soil layer.

As presented in Figure 6a, the numerical analysis (FLAC-3D) is conducted under an axisymmetric model, which adopts the unit-cell concept applied in the theoretical approach. The axisymmetric boundary conditions are adopted. The vertical boundary only allows for vertical displacement and the bottom boundary is fixed in any direction. Figures 6b and 6c are the profile of the pile with conical head and the equal section pile, respectively. The pile is simulated as a linear elastic material. Meanwhile, the soft soil and embankment fills are simulated as linearly elastic-perfectly plastic material with the Mohr-Coulomb failure criterion. The interface parameters between the pile and soil include normal stiffness $k_n$, shear stiffness $k_s$, cohesion $c$, and internal friction angle $\varphi_s$. $k_n$ and $k_s$ can be taken as ten times the stiffness of the hardest soil layer in the adjacent contact surface [35]. The value of $c$ and $\varphi_s$ are respectively equal to 0.8 times the cohesion and internal friction angle of the hardest soil layer [35], as shown in Equation (39). The material parameters and geometric parameters used in the model are listed in Table 1.

$$k_n = k_s = 10\max\left[\frac{K + \frac{4}{3}G}{\Delta z_{\min}}\right] \tag{39}$$

where $\Delta z_{\min}$ is the least grid size in this model, and $K$ and $G$ represent the bulk modulus and shear modulus of the soil respectively.

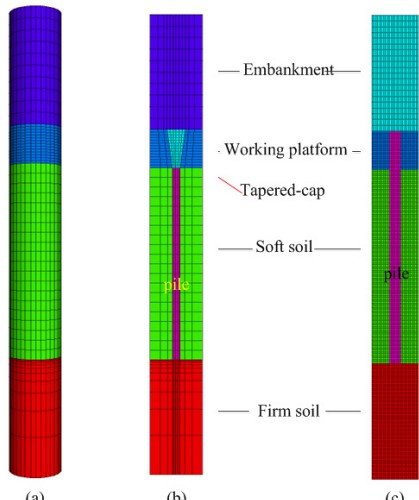

**Figure 6.** The numerical model: (**a**) the unit cell of the pile supported embankment, (**b**) the profile of the pile with a conical head, (**c**) the profile of the equal section pile.

**Table 1.** Material properties and geometric parameters used in the numerical model.

| Items | Embankment | Soft Soil | Firm Soil | Working Platform | Pile |
|---|---|---|---|---|---|
| Height (m) | 6 | 10 | 6 | 2 | 12 |
| Young's modulus (MPa) | 30 | 7 | 48 | 15 | 30000 |
| Poisson's ratio | 0.25 | 0.25 | 0.35 | 0.25 | 0.15 |
| Cohesion (kPa) | 0 | 15 | 30 | 10 | - |
| Friction angle (°) | 30 | 9 | 22 | 35 | - |
| Density (kg/m$^3$) | 2000 | 1750 | 1750 | 2000 | 2500 |

Figure 7 shows the variation of vertical stress within the inner cylinder and the outer hollow cylinder versus the height of the embankment. The curves of the proposed solution are much closer to the numerical simulation. The vertical stresses between the inner cylinder and the outer hollow cylinder are equal when $z/h$ is less than 0.625. It means that there is a plane of equal settlement at $z/h = 0.625$. Below the plane of equal settlement, the vertical stress gradually deviates from the gravity while the vertical stress of the outer hollow cylinder gradually decreases and that of the inner cylinder increases. The height of the equal settlement plane is 1.41 $s_n$. The result agrees with that obtained from the experiment of Cao et al. 2007 [36]. Thus, the proposed solution is feasible for predicting the soil arching effect for a pile-supported embankment with a conical head.

Figure 8 indicates the effect of $D$ on the pile–soil stress ratio ($n$) and the load sharing ratio ($n_e$). The pile–soil stress ratio decreases (Figure 8a) while the load sharing ratio is increased (Figure 8b) with the increase in $D$. The comparison indicates that the predicted result is highly close to the numerical result, where the relative error of the load sharing ratio floats within the range of 1.6%–9.6%. The relative error of the two curves increases with the value of $D$. It is because the pile–soil stress ratio is the ratio of the average vertical stress of the pile to that of the soft soil at the plane of the pile top. The larger $D$ is, the greater the deviation between the calculated value of the average stress and the actual one will be.

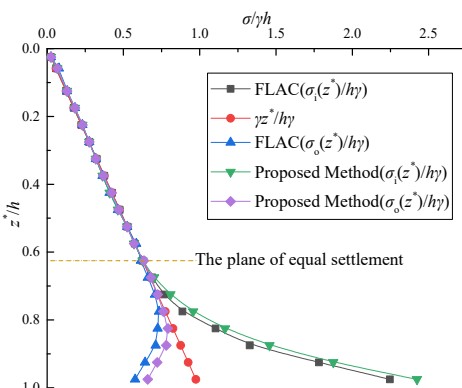

**Figure 7.** Vertical stress distribution of the inner cylinder and outer hollow cylinder in the embankment.

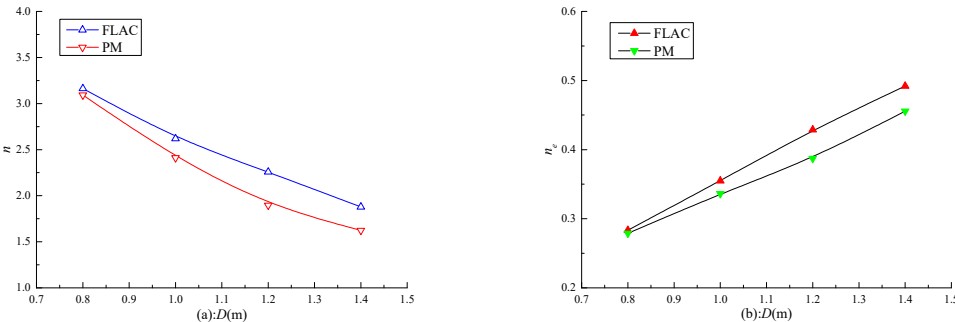

**Figure 8.** (**a**). The pile–soil stress ratio vs. *D*. (**b**) The load sharing ratio vs. *D*.

It is further to verify the feasibility of the proposed solution, but slight discrepancy is inevitable owing to the existence of assumptions. For instance, the embankment fill is divided into an inner cylinder and an outer hollow cylinder and the vertical stress and vertical deformation are deemed to be uniform.

### 3.2. Validation 2

The proposed solution can be used to evaluate not only the pile-supported embankment with the conical head, but also the pile-supported embankment with the equal-section pile. When the proposed solution is utilized to calculate the equal-section piles, the zone of the conical head is ignored and setting $L_0 = 0$. In this section, a case of equal-section pile-supported embankment is introduced to verify the proposed method. The literature includes two cases: Floating pile and End-bearing pile [23]. The proposed solution is applicable to the former, in which the pile is 20 m long and installed in a 25 m thick soft layer. Therefore, the pile toe is in the soft soil and has a 5 m underlying soft soil. Chen built an axisymmetric model in PLAXIS (a finite element method), which simulated the unit-cell concept utilized in the theoretical solution. The pile and caps were modeled as linearly elastic materials. The soft soil and the embankment fill were modeled as linearly elastic-perfectly plastic materials. All parameters used in the calculation are summarized in Table 2.

**Table 2.** Material properties and geometric parameters used in validation 2.

| Items | Embankment | Soft Soil | Firm Soil | Cap | Pile |
|---|---|---|---|---|---|
| Height (m) | 4 | 25 | 6 | 0.35 | 20 |
| Friction angle (°) | 30 | | | | |
| Effective friction angle (°) | | 9 | 22 | | |
| Cohesion (kPa) | 0 | | | | |
| Effective Cohesion (kPa) | | 15 | 30 | | |
| Young's modulus (MPa) | 30 | | | 35 | 35 |
| Constrained modulus (MPa) | | 2.2 | 15 | | |
| Poisson's ratio | 0.25 | 0.35 | 0.35 | 0.15 | 0.15 |
| Density (kg/m$^3$) | 2000 | | | 2500 | 2500 |
| Saturated unit weight (kN/m$^3$) | | 17.5 | 18 | | |
| Pile spacing (m) | | | | 1 | 2.5 |
| Diameter (m) | | | | 1.13 | 0.4 |

The load sharing ratios ($n_e$) predicted by the proposed method, the literature method, and the FEM are 65.1%, 74%, and 66.7%, respectively, and the pile–soil stress ratios ($n$) are 9.792, 14.942, and 10.515, respectively. The comparison results further prove the rationality of this method, and the proposed method can be applied for the case of equal-section pile and variable-section pile in a pile-supported embankment.

## 4. Parametric Studies

Since the main research object is the load transfer mechanism of piles with pyramidal caps, the height of embankment ($h$) and the influence radius of single pile ($r_e$) are the main geometric parameters affecting the settlement and pile–soil stress ratio. Therefore, the embankment height ($h$), the maximum diameter of the conical head ($D$), and the effective reinforcement radius of pile ($r_e$) are selected for parameter analysis. The parameter analysis scheme were summarized in Table 3 and the results were displayed in Figures 9–16. These values were used throughout unless otherwise specified.

**Table 3.** Material consumption of the two types of pile and the settlement value.

| Pile Diameter (m) | | Pile Volume (m$^3$) | | Settlement (mm) | | Percentage of Material Consumption |
|---|---|---|---|---|---|---|
| $D$ | $d$ | Pile with the Conical Head | Equal Section Pile | Pile with the Conical Head | Equal Section Pile | |
| 0.6 | 0.4 | 1.65 | 3.39 | 65.19 | 65.74 | 51.30% |
| 0.8 | 0.4 | 1.84 | 6.03 | 62.46 | 63.52 | 69.40% |
| 1 | 0.4 | 2.07 | 9.42 | 60.71 | 62.12 | 78% |
| 1.2 | 0.4 | 2.34 | 13.6 | 58.71 | 61.23 | 82.70% |
| 1.4 | 0.4 | 2.66 | 18.5 | 56.37 | 61.39 | 85.60% |

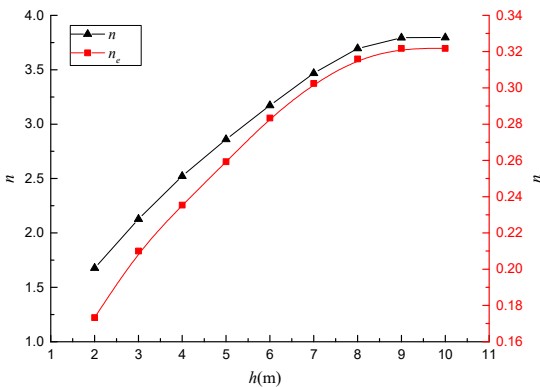

**Figure 9.** Pile–soil stress ratio and the load sharing ratio vs. the height of the embankment.

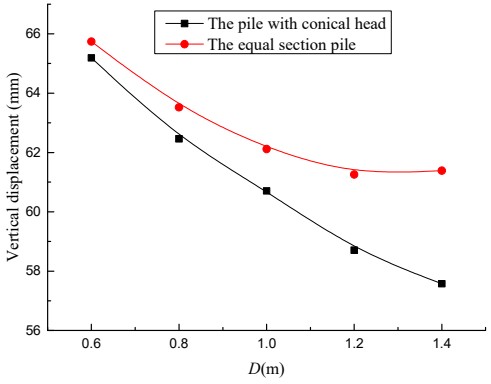

**Figure 10.** The vertical displacement of embankment surface vs. *D.*

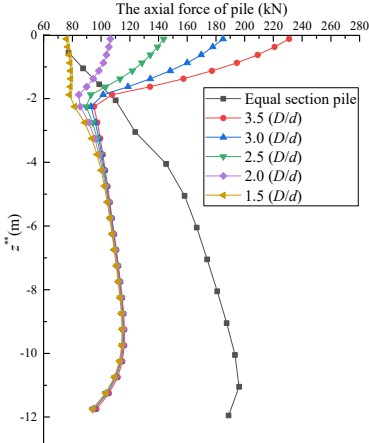

**Figure 11.** The axial force on the pile.

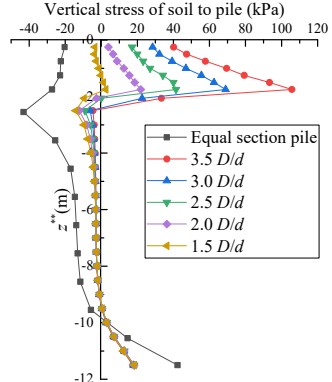

**Figure 12.** The vertical stress curve of the soil to pile.

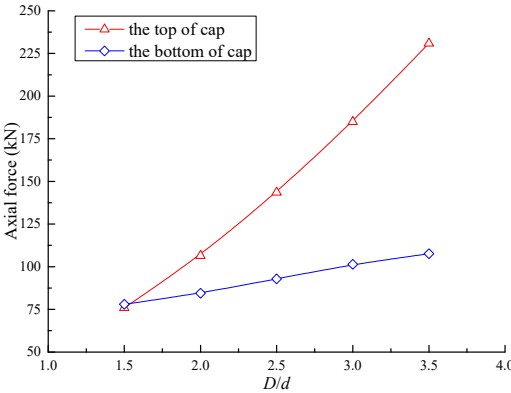

**Figure 13.** The axial force at the top and the bottom of the conical head vs. *D/d*.

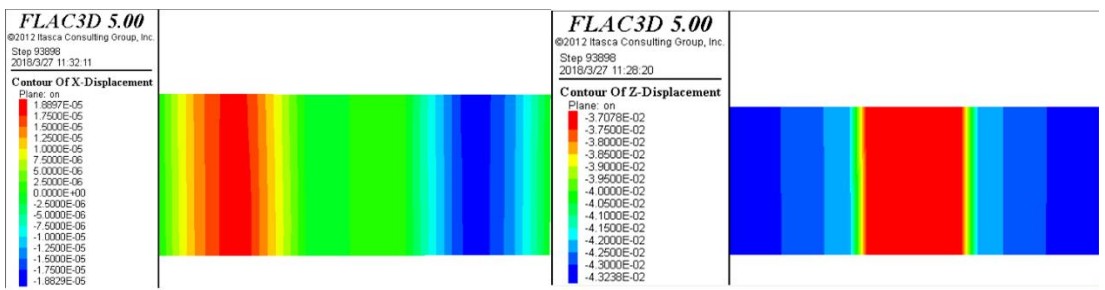

**Figure 14.** The horizontal and vertical displacement nephogram.

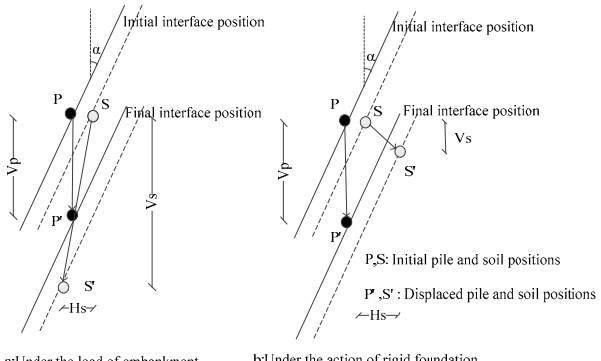

**Figure 15.** The relative displacement of the pile and soil.

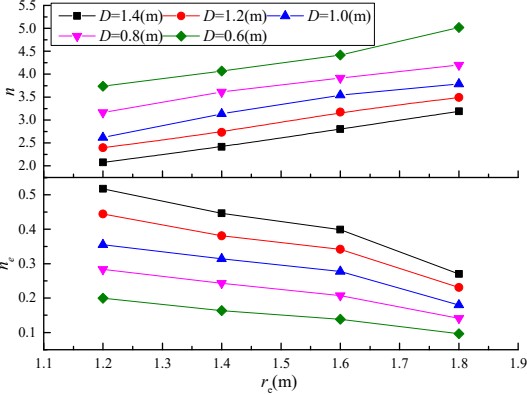

**Figure 16.** The pile–soil stress ratio and the load sharing ratio vs. $r_e$.

### 4.1. Effect of the Embankment Height (h)

Figure 9 describe the variation of $n$ and $n_e$ with the increase in the embankment height when $r_e = 1.2$ m. The pile–soil stress ratio ($n$) and the load sharing ratio ($n_e$) increase with the height of the embankment and are followed by a relatively stable state. Specifically speaking, more than 68% of the embankment load is carried by the subsoil when $h = 10$ m and $m = 11.1\%$. The same conclusion has been achieved by Zhuang [37]. The results indicate that the foundation soil occupies the main position in bearing the load of the embankment. In addition, when $h > 1.41\ s_n$, the embankment has the geometrical condition to form a full soil arch. The embankment loads above the plane of equal settlement are not all transferred to the pile, but are assigned to the pile and soil according to the load sharing ratio. The same conclusion has been reached by Girout [38].

### 4.2. Effect of the Maximum Diameter of the Conical Head (D)

Figure 10 shows that the vertical displacement of the embankment surface changes as $D$ increases from 0.6 m to 1.4 m in the case $d = 0.4$m (Minimum pile diameter) and $r_e = 1.2$ m. As is anticipated, the settlement of the embankment surface decreases with an increase in $D$, but when the diameter exceeds 1 m, the curve of settlement for the equal section pile flattens out gradually. The settlement of the embankment reinforced by the pile with conical head is less than that of the equal section pile when the maximum diameter of the conical head is equal to the diameter of the equal section pile. The difference of two curves become increasingly obvious as $D$ increases, which further testifies to the superiority of the pile with a conical head as $D$ grows. Table 3 shows the material consumption of two types of piles. The comparative result indicates that a pile with a conical head consumes less material and results in less settlement. It can be observed that the piles with conical heads have better working performance and practicability.

Figure 11 shows the variation of the axial force of pile with $D/d$. The axial force of the equal section pile increases with depth on the whole, and the maximum value is 196 kN at the neutral point (11 m below the ground surface). However, the neutral point of the pile with a conical head is located at 9 m below the ground surface with a maximum axial force of 116 kN. Below the conical head, the variation tendency of the axial force is similar and the difference is small although $D/d$ differs. In the design of pile supported embankments, the pile diameter is determined by the maximum axial force and the construction machinery. The area replacement rate is a constant for the equal section pile, which results in a high area placement rate except for the neutral point. Compared with the equal section pile, the axial force below the conical head varies within a relatively small range, which is consistent with the principle that the greater the axial force is, the larger the cross section of the pile.

As shown in Figure 11, the axial force at the pile top increases with the value of $D/d$. While the axial force gradually decreases as $z^{**}$ goes deeper within the conical head zone, and the bigger the $D/d$ is, the faster the axial force decreases. Obviously, the axial force variation trend of the two types of pile stands opposite within the conical head zone. The conical head converts a portion of the vertical load into a horizontal load and transmits it into the uppermost crust (with relatively high plasticity, a water content slightly lower than the plastic limit, and a relatively high over consolidation ratio (OCR)) or artificial working platform. This process contributes to improving the bearing capacity of pile supported embankments. When $D/d = 1.5$, the maximum diameter of the conical head and the diameter of the equal section pile both are 0.6 m. The axial force at the top of the two types of piles is 76 kN. Within the conical head zone, the axial force of pile with a conical head remains unchanged while that of the equal section pile climbs to 110 kN. It can be concluded that the vertical interaction force between the pile and soil is zero within the conical head zone when $D/d = 1.5$. The reason will be unfolded below.

Figure. 12 shows the vertical interaction stress ($F(z^{**})$) versus $z^{**}$. For the pile with a conical head, as $D/d$ increases, the vertical interaction stress changes from negative to positive within the conical head zone. This does not mean that the conical head shifts downward relative to the soil. As shown in

Figure 5, the vertical stress of the soil to the pile is composed of two parts: the vertical component of friction and the vertical component of the normal force.

Figure 13 illustrates the axial force at the top and bottom of the conical head. As $D/d$ grows, the axial force at the bottom of the conical head increases slowly, while that at the top of the conical head increases rapidly. It can be seen that the conical head can effectively reduce the axial force of the pile and the larger the $D/d$ is, the more obvious the effect will be.

To judge the relative displacement direction of pile and soil, the "plain function" in the FLAC-3D is applied to slice the horizontal and vertical displacement nephogram along horizontal direction at one meter below the pile top, as is shown in Figure 14. When the settlement of soil is greater than that of the conical head, the soil would detach from the interface while the upper soil would move down to fill the gap. In this way, pressure and friction are generated on the interface. The relative displacement of the pile against the soil under the embankment is shown in Figure 15a and one under rigid foundation is shown in Figure 15b [39]. The above mentioned analysis elaborates that under the load of the embankment, the friction direction of soil against pile is downward at the conical head zone and the vertical stress of pile–soil interaction can be formulated as follows:

$$F(z^{**}) = F_{Nz^{**}}(z^{**}) - F_{\tau z^{**}}(z^{**}) = F_N(z^{**}) \sin \alpha - F_\tau(z^{**}) \cos \alpha \tag{40}$$

It can be seen from Equation (39) that the vertical stress of soil against pile increases with $\alpha$ (the cone angle of the conical head), which is consistent with the conclusion made in the numerical simulation. From Equation (14), we can see that when $\alpha$ is constant, the vertical stress of the soil on the pile is only related to the soil stress $P_s(z^{**})$. This is consistent with the result shown in Figure 12, that is, the vertical stress of soil to pile changes linearly in the conical head zone. The comparative analysis indicates that a pile with a conical head can effectively reduce negative friction, which is in agreement with the conclusion of Sawaguchi et al [14] on a model test. A pile with a conical head can effectively improve the position of the neutral point, reduce negative friction, and improve bearing capacity.

### 4.3. Effect of the Effective Reinforcement Radius of the Pile($r_e$)

In this section, the embankment height is 6 m and the effective radius ranges from 1.2 m to 1.8m. Figure 16 reveals that the pile–soil stress ratio positively correlates with $r_e$ while the load sharing ratio negatively correlates with $r_e$. Moreover, the area replacement rate ($m = 0.111$) is the same under the two conditions where $r_e = 1.2$ m, $D = 0.8$ m, and $r_e = 1.8$ m, $D = 1.2$ m. In both cases, the pile–soil stress ratios are respectively 3.16 and 3.49, and the load sharing ratios are 28.3% and 23.1% accordingly. A comparative result indicates that the latter should be applied in the design of a piled-supported embankment.

### 5. Conclusions

In this paper, a theoretical method is proposed to analyze the pile-supported embankment with a conical head. In the derivations, it is assumed that the piles, embankment fill, and soft soil only have vertical deformation. The proposed method can determine the plane of equal settlement, the pile–soil stress ratio ($n$), the load sharing ratio ($n_e$), and the friction. The method has been proved to be feasible since it is verified by comparison with the numerical results and a theoretical study. There are three lightspots in the method: First, the method can be used to evaluate the pile-supported embankment with a conical head; Second, the improved frictional soil arching model can consider the effect of pile–soil interaction on the development of soil arching; Third, the critical height of the soil arch is not pre-assumed, but is calculated under the deformation continuity condition. The main findings can be summarized as follows:

1. The conical head converts a portion of the vertical load into a horizontal load and transmits it into the uppermost crust or artificial working platform, which helps to elevate the position of the neutral point and reduce the settlement of the embankment;

2.     When the height of the embankment is greater than that of the plane of equal settlement, increasing the height of the embankment has little benefit to the load sharing ratio;

3.     Not all of the fill load above the plane of equal settlement is transmitted to the pile through the soil arch but is assigned to the pile and soil according to the load sharing ratio.

In conclusion, under the theoretical method in this paper, a more effective pile-supported embankment with a conical head can be designed by capturing the load transfer mechanics of all the components in a more accurate way.

**Author Contributions:** Conceptualization, C.Z.; Funding acquisition, M.Z.; Methodology, C.Z.; Software, Z.X.; Validation, C.Z.; Writing—original draft, C.Z.; Writing—review & editing, C.Z. and S.Z.

**Funding:** This research is a part of the work funded by grants from the National Natural Science Foundation of China (Nos. 51,478,178 and 51608540).

**Acknowledgments:** This research is a part of the work funded by grants from the National Natural Science Foundation of China (Nos. 51478178 and 51608540), which made the work presented in this paper possible. The authors would like to thank professor the professor Minghua Zhao for his help in this research.

**Conflicts of Interest:** The authors declare no conflict of interest.

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
