# Peer review of "A Theoretical Solution for Pile-Supported Embankment with a Conical Pile-Head"

_applsci, doi:10.3390/app9132658_

Round 1

Reviewer 1 Report

The paper entitled "A Theoretical Solution for Pile Supported Embankment with a Conical Pile Head" is a good and interesting research, based on the pile supported embankment with a conical pile head. The paper proposes a theoretical solution, which incorporates all the load transfer mechanisms, namely the soil arching effect, the interaction between pile and soil and the support from the sub stratum.

- The paper can be published after some corrections.

- Line 55: “Daniel [15]” is probably “Dias [15]”;

- Line 102: “Zhao and Chem [23, 29]” is better “Chen et al. [23]” and “Zhao et al. [29]”;

- Line 212: “Berrum method [30]” is perhaps “Randolph method [30]”;

- Table 2: “Unit weight (kN/m3)” is perhaps “Unit weight (kg/m3)”;

- Line 361: For a better understanding of the text, it would be appropriate to remember the meaning of the term “d”;

- Line 427: For a better understanding of the text, it would be appropriate to remember the meaning of the term “a” (Line 181 and 182).

Author Response

Response to Reviewer 1 Comments

Thank you for your valuable advice. We have revised the manuscript according to your opinions. Detailed corrections and improvements are listed below.

Point 1:. “Daniel [15]” is probably “Dias [15]”;

Response 1: According to the reviewer's opinion, the manuscript has been revised. The details can be found in Line 55 of the revised version.

Point 2:. Line 102: “Zhao and Chen [23, 29]” is better “Chen et al. [23]” and “Zhao et al. [29]”;

Response 2: According to the reviewer's opinion, the manuscript has been revised. The details can be found in Line 102 of the revised version.

Point 3:. Line 212: “Berrum method [30]” is perhaps “Randolph method [30]”;

Response 3: According to the reviewer's opinion, the manuscript has been revised. The details can be found in Line 212 of the revised version.

Point 4:. Table 2: “Unit weight (kN/m3)” is perhaps “Unit weight (kg/m3)”;

Response 4: The author read the manuscript carefully. The parameter should be density and the unit is kg/m3. It has been revised. The details can be found in Table 2 of the revised version.

Point 5:. Line 361: For a better understanding of the text, it would be appropriate to remember the meaning of the term “d”;

Response 5: According to the reviewer's opinion, the manuscript has been revised. The details can be found in Line 367 of the revised version.

Point 6:. Line 427: For a better understanding of the text, it would be appropriate to remember the meaning of the term “a” (Line 181 and 182).

Response 6: According to the reviewer's opinion, the manuscript has been revised. The details can be found in Line 434 of the revised version.

Reviewer 2 Report

The manuscript presents a theoretical approach to evaluate the performance of pile-supported embankments with a conical pile-head, which receives little attention. In the current study, an improved cylindrical unit cell model is proposed and validated by two methods, including a numerical analysis based on the finite difference approach (FLAC 3D) and an analytical solution. This paper is interesting and well organized, however, some minor revisions have to be made before publication.

Technical issues

1. As a scientific paper, the authors should explain the cases the proposed model could apply and clarify its limitations.

2. The authors should provide more details on the numerical models.

3. The conclusion should be rewritten and covers primary highlights.

4. The parametric study is restricted for parameters re, h and D, why? Are there any other factors should be taken into consideration?

Editorial issues

1. Line15: “the interaction between pile and soil” is suggested to rewritten as “the pile-soil interaction”

2. line 27 and line76: The term “pile friction” is not uniform. (the pile friction / the side friction force)

3. line38: The term “variable section pile” (a circular pile with a conical head) should be annotated as the pile with a cone cap.

4. line60: the ‘degree of compaction’ should be ‘compactness’

5. Line104 and Line109: The spacing and net spacing between adjacent piles is respectively denoted as s and sn. To prevent confusion, spacing could be called axis to axis spacing.

6. line 143: The parameter S0 is not defined.

7. line153, Parentheses should not be included in subscripts, please correct.

8. line122 and line 210, subscripts should be added to distinguish the active earth pressure coefficients in fill and in soft soil, respectively.

Author Response

Response to Reviewer 2 Comments

Thank you for your valuable advice. We have revised the manuscript according to your opinions. Detailed corrections and improvements are listed below.

Technical issues

Point 1: 1. As a scientific paper, the authors should explain the cases the proposed model could apply and clarify its limitations.

Response 1: As stated by the reviewer, the theoretical method in this paper is not applicable to all pile-supported embankment, and it is not applicable when the embankment filling height is low. The author has supplemented this in the article. The details can be found in Line 134 of the revised version.

Point 2: The authors should provide more details on the numerical model.

Response 2: According to the reviewer's opinion, the author further described the numerical model in detail. The details can be found in Lines 281-284 of the revised version.

Point 3: The conclusion should be rewritten and covers primary highlights.

Response 3: According to the reviewer's opinions, the author has modified the conclusion and summarized the main conclusions of this part. The details can be found in Lines 463-469 of the revised version.

Point 4: The parametric study is restricted for parameters re, h and D, why? Are there any other factors should be taken into consideration? .

Response 4: The main research object of this paper is the load transfer mechanism of piles with pyramidal caps. The height of embankment filling and the influence radius of single pile are the geometric parameters that mainly affect the settlement and stress ratio of pile-supported embankment. Therefore, re, h and D are selected in this paper for parameter analysis. In the part of parameter analysis, the author adds the description of parameter selection. The details can be found in Lines 345-349 of the revised version.

 Editorial issues

Point 1:. Line15: “the interaction between pile and soil” is suggested to rewritten as “the pile-soil interaction”

Response 1: According to the reviewer's comments, the author changed the term " the interaction between pile and soil " to " the pile-soil interaction ", and checked the full text. The details can be found in Line 20 of the revised version

Point 2: line 27 and line76: The term “pile friction” is not uniform. (the pile friction / the side friction force)

Response 2: According to the expert opinions of the reviewer, The term " the pile shaft friction" is used to replace the term " the pile friction " and " the side friction force ". The details can be found in Lines 26 and 81 of the revised version

Point 3:. line38: The term “variable section pile” should be annotated as the pile with a cone cap.

Response 3: According to the expert opinions of the reviewer, The term" variable section pile " generally refers to the pile with the change of section, and the author has modified " variable section pile " to " the pile with a cone cap (a type of variable section pile)". The details can be found in Line 43 of the revised version

Point 4:. line60: the ‘degree of compaction’ should be ‘compactness’

Response 4: According to the comments of reviewers, the author has made modifications in the corresponding places in the paper about the term of "compactness". The details can be found in Line 66 of the revised version

Point 5:. Line104 and Line109: The spacing and net spacing between adjacent piles is respectively denoted as s and sn. To prevent confusion, spacing could be called axis to axis spacing.

Response 5: According to the reviewer's opinion. To avoid confusion, the author changed the term "spacing" to "axis to axis spacing". The details can be found in Lines 110 and 115 of the revised version.

Point 6:. line 143: The parameter S0 is not defined.

Response 6: According to the opinions of reviewers, the author supplemented the definition of parameter " S0". The details can be found in Line 152 of the revised version.

Point 7:. line153, Parentheses should not be included in subscripts, please correct.

Response 7: The author has corrected the write errors in the manuscript. The details can be found in Line 161 of the revised version.

Point 8:. line122 and line 210, subscripts should be added to distinguish the active earth pressure coefficients in fill and in soft soil, respectively.

Response 8: According to the reviewer's opinion, the author adds a subscript to the active earth pressure coefficient to distinguish the active earth pressure coefficient in embankment and the active earth pressure in soft soil. kae and kas stand for the active earth pressure coefficient in embankment and the active earth pressure coefficient in soft soil, respectively. The details can be found in Lines 128 and 218 of the revised version.